

# Changes in the composition of invertebrate assemblages from wave-exposed intertidal mussel stands along the Nova Scotia coast, Canada

Ricardo A. Scrosati[1] and Julius A. Ellrich[2]

[1] Department of Biology, St. Francis Xavier University, Antigonish, Nova Scotia, Canada
[2] Alfred Wegener Institute Helmholtz Centre for Polar and Marine Research, Helgoland, Germany

## ABSTRACT

Rocky intertidal habitats occur worldwide and are mainly characterized by primary space holders such as seaweeds and sessile invertebrates. Some of these organisms are foundation species, as they can form structurally complex stands that host many small invertebrates. The abundance of primary space holders is known to vary along coastlines driven directly or indirectly by environmental variation. However, it is less clear if the invertebrate assemblages associated to a foundation species may remain relatively unchanged along coastlines, as similar stands of a foundation species can generate similar microclimates. We examined this question using abundance data for invertebrate species found in mussel stands of a similar structure in wave-exposed rocky habitats at mid-intertidal elevations along the Atlantic coast of Nova Scotia (Canada). While the most abundant invertebrate species were found at three locations spanning 315 km of coastline, species composition (a combined measure of species identity and their relative abundance) differed significantly among the locations. One of the species explaining the highest amount of variation among locations (a barnacle) exhibited potential signs of bottom-up regulation involving pelagic food supply, suggesting benthic–pelagic coupling. The abundance of the species that explained the highest amount of variation (an oligochaete) was positively related to the abundance of their predators (mites), further suggesting bottom-up forcing in these communities. Overall, we conclude that species assemblages associated to structurally similar stands of a foundation species can show clear changes in species composition at a regional scale.

## INTRODUCTION

Biogeographic regions are characterized by particular species pools that result from unique evolutionary histories (*Carstensen et al., 2013*; *Cox, Moore & Ladle, 2016*). Within regions, the relative abundance of species often changes across locations in response to various ecological factors (*Vellend, 2016*). Rocky intertidal systems, which are those delimited by the highest and lowest tides on marine rocky shores, are no exception. These habitats occur all over the world and host unique biotas owing to the daily alternation of submergence and emersion periods due to tides. The organisms that dominate these systems are primary space

Corresponding author
Ricardo A. Scrosati, rscrosat@stfx.ca

holders such as seaweeds and sessile invertebrates, followed by their motile consumers. Variation in the relative abundance of these species along coastlines has been documented for several biogeographic regions, many studies exploring the underlying ecological drivers (*Bustamante, Branch & Eekhout, 1997*; *Broitman et al., 2011*; *Bryson, Trussell & Ewanchuk, 2014*; *Griffiths & Waller, 2016*; *Hawkins et al., 2019*; *Ibanez-Erquiaga et al., 2018*; *Menge et al., 2019*; *Palomo et al., 2019*; *Ishida et al., 2021*; *Thyrring et al., 2021*; *Scrosati et al., 2022*; *Trott, 2022*; *Gilson & McQuaid, 2023*; *Pardal et al., 2023*).

Rocky intertidal systems also host many small invertebrate species that often remain undetected because of their size and the places where they occur. Such is the case of the invertebrates that live in stands of intertidal foundation species (*Lafferty & Suchanek, 2016*). Foundation species are primary space holders whose body structures create complex habitats that provide shelter from abiotic and biotic stresses (*Stachowicz, 2001*; *Altieri & van de Koppel, 2014*; *Ellison, 2019*). In rocky intertidal systems, common foundation species are seaweeds, tunicates, vermetid snails, and mussels. Dense stands of these organisms limit thermal and desiccation stress during low tides and hydrodynamic stress during high tides, so they often host tens of small invertebrate species (*Cerda & Castilla, 2001*; *Borthagaray & Carranza, 2007*; *Silliman et al., 2011*; *Donnarumma et al., 2018*; *Catalán et al., 2021*; *Catalán et al., 2023*). As these invertebrate assemblages can enrich intertidal biodiversity considerably, efforts to understand their ecology are increasing.

A fundamental question is how geographically variable the composition of species assemblages associated to a foundation species can be. It is well known that, through environmental filtering and influences on interspecific interactions, abiotic variation along rocky coastlines affects the relative abundance of primary space holders (*Menge & Branch, 2001*; *Sanford, 2014*). However, depending on their structural and functional properties, foundation species create specific conditions within their stands that favour particular sets of associated species (*Cameron, Scrosati & Valdivia, 2024a*). In theory, then, the species assemblages living in stands of a particular foundation species might depend little on external factors as long as stands of that foundation species remain similar along coastlines. This notion has been supported by a study done on the NE Pacific coast that showed that the thermal buffering conferred by dense intertidal stands of mussels and seaweeds during low tides can be more important for associated species than latitudinal thermal changes (*Jurgens & Gaylord, 2018*). Consideration of overall environmental variation, however, has yielded opposite results, as large-scale environmental changes along the South African coast influence the fauna living in stands of a mussel species more than mussel stand structure (*Cole & McQuaid, 2011*). Intermediate results have been obtained for other systems, as communities associated to macroalgal foundation species on the SW Pacific coast depend on characteristics of the algal stands as well as on external environmental conditions (*Lloyd et al., 2020*). Clearly, relative to primary space holders, there is much to learn about the drivers that structure the communities associated to rocky-intertidal foundation species along coastlines.

In this study, we examine this issue using data on the abundance of invertebrates found in mussel stands from wave-exposed rocky intertidal habitats along the Atlantic coast of Nova Scotia, Canada. The overall spatial coverage of mussel beds in such habitats varies

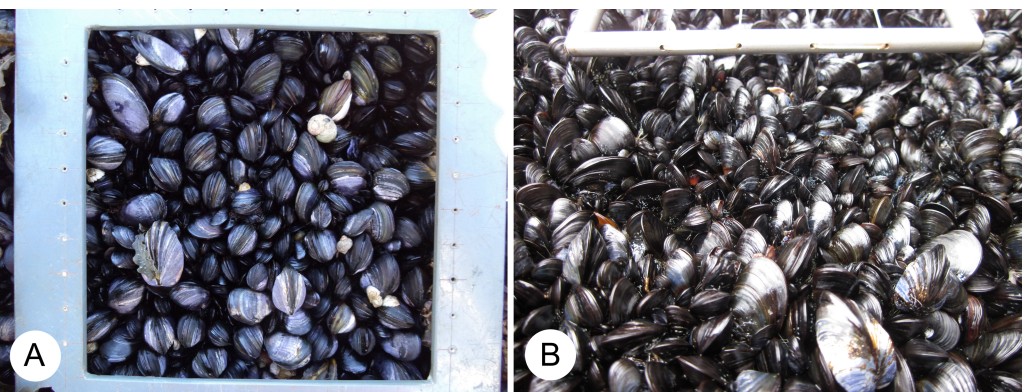

**Figure 1** **Wave-exposed intertidal mussel stands.** Typical mussel stands from wave-exposed rocky intertidal habitats photographed at low tide on the (A) southern and (B) central open Atlantic coast of mainland Nova Scotia, Canada. The inner frame of the sampling quadrats shown in these photographs measures 10 cm x 10 cm. Photo credits: Ricardo A. Scrosati.

along this coast in relation to alongshore changes in environmental conditions (*Scrosati & Ellrich, 2018*; *Scrosati, Freeman & Ellrich, 2020*; *Scrosati et al., 2022*). However, in these wave-exposed habitats, mussels consistently occur as predominantly monolayered stands of densely packed small individuals (*Tam & Scrosati, 2011*; *Tam & Scrosati, 2014*; Fig. 1). Thus, we surveyed mussel patches of an equal area from those habitats to test if the associated invertebrate communities are similar along the coast or if they vary possibly in relation to external environmental conditions.

## MATERIAL AND METHODS

To address this goal, we used data that were originally collected to describe the invertebrate assemblages occurring in rocky-intertidal mussel stands in Nova Scotia (*Arribas et al., 2014*). The full dataset is available from an associated data paper (*Scrosati, Arribas & Donnarumma, 2020*). For the present study, we used the data collected only in wave-exposed habitats because of their convenient spatial coverage for our goals. The following paragraph summarizes the sampling design.

Between early September and early October 2012, three wave-exposed locations spanning 315 km of the Atlantic coast of Nova Scotia (Fig. 2) were sampled during low tides. These are cold-temperate locations located in the same hydrographic subregion (Nova Scotia) of the NW Atlantic cold-temperate biogeographic region (*Mathieson, Penniman & Harris, 1991*; *Spalding et al., 2007*). The surveyed habitats were middle intertidal elevations, have bedrock as a substrate, and face the open ocean directly. The locations were, from north to south, Tor Bay Provincial Park (45.182894, -61.353258; TB hereafter), Crystal Crescent Beach (44.447372, -63.622214; CC), and Kejimkujik National Park (43.818614, -64.834747; KE). Values of daily maximum water velocity (a proxy for wave exposure) in wave-exposed rocky intertidal habitats on this coast average 8 m s$^{-1}$ (*Scrosati & Heaven, 2007*), with peaks of 12 m s$^{-1}$ (*Hunt & Scheibling, 2001*). Two species of blue mussel (*Mytilus edulis* and *M. trossulus*) form extensive stands in wave-exposed rocky intertidal habitats on

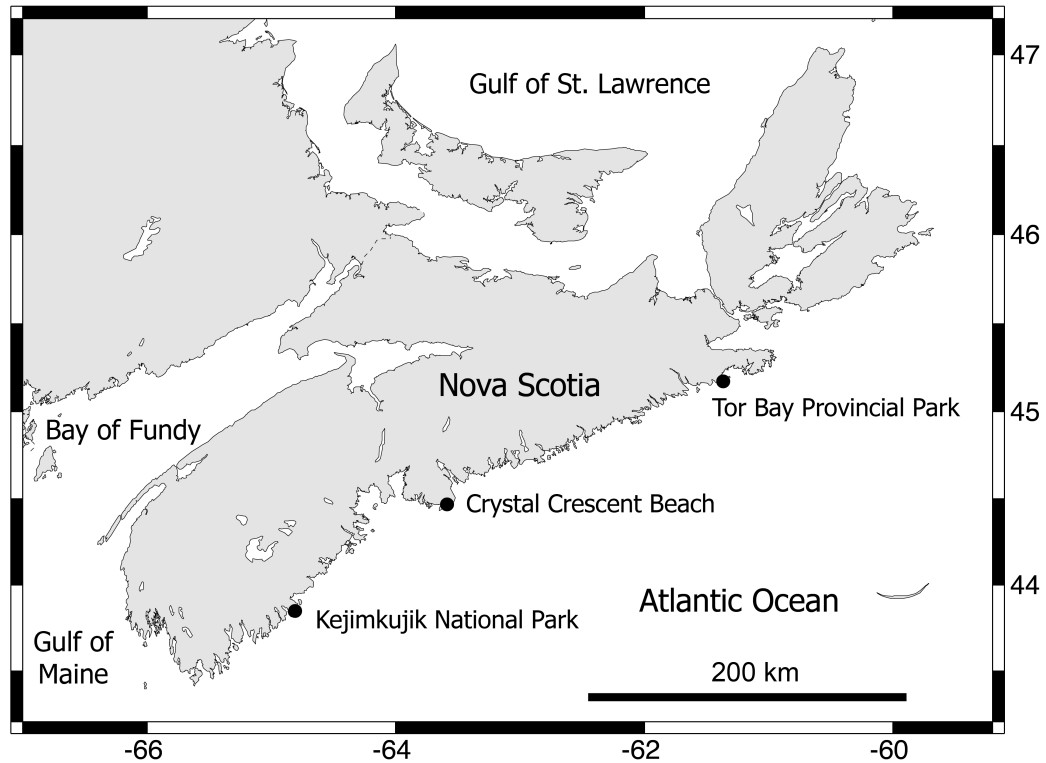

**Figure 2** **Map of the Nova Scotia coast.** This map indicates the wave-exposed rocky intertidal locations that were surveyed for this study. Map source credit: *Scrosati & Ellrich (2020)*.

this coast. Both species are morphologically similar (*Innes & Bates, 1999*) and occur in mixed stands in these habitats, with a predominance of *M. trossulus* (80–85%) over *M. edulis* according to genetic analyses (*Hunt & Scheibling, 1996*; *Tam & Scrosati, 2011*). In these wave-exposed habitats, mussels are consistently small (mean length <1 cm (*Tam & Scrosati, 2011*; *Tam & Scrosati, 2014*)) due to wave-imposed size limits (*Carrington et al., 2009*) and form predominantly monolayered stands with high densities of individuals (Fig. 1). At each studied location, on wave-exposed areas of the substrate that were fully covered by mussels (Fig. 1), the mussels and all associated invertebrates were collected from 15 random patches measuring 100 cm$^2$ in area. The associated invertebrates were identified to the lowest possible taxonomic level and their abundance was measured for each patch.

Using those data, we tested if the composition of the associated communities (combined measure of species identity and their relative abundance) differed among locations through a permutational multivariate analysis of variance (PERMANOVA) based on Bray-Curtis distances between patches and 9,999 permutations for significance testing. To visualize the compositional differences between locations, we generated a nonmetric multidimensional scaling (NMDS) ordination of patches based on Bray-Curtis similarity scores. To identify the species that contributed the most to the differences in composition between locations, we ran analyses of similarity percentages (SIMPER). We did these analyses with PRIMER

6.1.18 plus PERMANOVA+ 1.0.8 software (*Anderson, Gorley & Clarke, 2008*; *Clarke et al., 2014*).

As species composition in mussel beds differed among locations and barnacle (*Semibalanus balanoides*) abundance explained much of that variation (see Results), we investigated if coastal food supply for barnacles was related to such biogeographic patterns. Barnacles are sessile organisms and a key step in their life cycle is recruitment (the appearance of new organisms on a substrate after larval settlement and metamorphosis). On our coast, barnacle recruitment occurs in the spring and is a good predictor of adult abundances later in the year (*Scrosati & Ellrich, 2018*). As data on barnacle recruitment within mussel stands were unavailable, we searched for differences among our locations in known important drivers of barnacle recruitment. One of such drivers is food supply for the larvae, which is related to larval performance and ultimately recruitment (*Vargas, Manríquez & Navarrete, 2006*). Barnacle larvae are pelagic and, before reaching the last (settling) stage, go over various nauplius stages that feed on phytoplankton for some weeks (5 to 6 for *S. balanoides*; *Bousfield, 1954*; *Bouchard & Aiken, 2012*). A proxy for phytoplankton abundance is the concentration of chlorophyll-a (chl-a hereafter) in seawater, which is regularly measured across the globe by satellites (*Mazzuco et al., 2015*; *Lara et al., 2016*). As most barnacle recruits appear by mid-May starting after the first week of May on our coast (*Scrosati & Holt, 2021*), most nauplius larvae are inferred to be in the water in April and the first half of May. Therefore, we looked for chl-a data for April and the first half of May 2012. Specifically, we obtained MODIS-Aqua satellite data on chl-a for the 4-km-$\times$-4-km cells that include our three study locations, retrieving the data from the Ocean Color website of the National Aeronautics and Space Administration (*NASA, 2024a*) using their SeaDAS software (*NASA, 2024b*).

## RESULTS

A total of 36 invertebrate taxa were identified in mussel stands from wave-exposed rocky habitats at mid-intertidal elevations on the Atlantic coast of Nova Scotia (Table 1). Twenty-four of these taxa were identified at the species level, while each of the 12 taxa identified above the species level likely represented one species, as indicated by consistent morphologies across the encountered individuals. Thus, we describe the findings of this study in terms of species metrics.

Sixteen species were found at all three studied locations, 13 of which were the species with the highest mean overall abundance (Table 1). Despite this commonality, the species composition of the invertebrate assemblages differed significantly among the locations (PERMANOVA's pseudo-$F = 7.199$, $p < 0.0001$; Fig. 3). Pairwise differences between locations were also significant, as the $p$ value was 0.0003 for the KE *vs.* TB comparison, 0.0001 for the KE *vs.* CC comparison, and 0.0018 for the TB *vs.* CC comparison. The variation in species composition explained by locations in the full model amounted to nearly a third of the total observed variation. The differences in species composition among locations were explained by abundance differences in the abundant species that were common to all three locations and by species with low average abundances being present only at certain subsets of locations (Table 1).

**Table 1  List of invertebrate taxa found in intertidal mussel stands from Nova Scotia ordered by their overall mean abundance (individuals dm⁻²).** The table also provides values of mean abundance for each of the three surveyed locations (TB, CC, and KE) and the percent contribution of each species to the dissimilarity in species composition between locations according to SIMPER tests.

| Scientific name | Common name | Overall mean abundance | Mean abundance at TB | Mean abundance at CC | Mean abundance at KE | % contribution KE *vs.* CC | % contribution KE *vs.* TB | % contribution CC *vs.* TB |
|---|---|---|---|---|---|---|---|---|
| Tubificidae (one species) | Oligochaete | 84.27 | 96 | 87.73 | 69.07 | 31.1 | 36.1 | 42.0 |
| *Semibalanus balanoides* | Barnacle | 19.82 | 14 | 3.67 | 41.8 | 29.7 | 20.2 | 8.6 |
| Nematoda (one species) | Nematode worm | 6.73 | 4.53 | 14.13 | 1.53 | 9.9 | 3.4 | 9.3 |
| Halacaridae (species 1) | Mite | 6.24 | 15.13 | 0.8 | 2.8 | 2.7 | 9.1 | 9.8 |
| *Littorina saxatilis* | Periwinkle | 4.96 | 3.53 | 3.53 | 7.8 | 4.8 | 3.8 | 2.3 |
| *Lasaea adansoni* | Bivalve mollusc | 4 | 8.33 | 1.07 | 2.6 | 2.0 | 5.9 | 5.8 |
| *Amphiporus angulatus* | Nemertean worm | 4 | 6.53 | 2.67 | 2.8 | 2.7 | 3.8 | 3.7 |
| Chironomidae (one species) | Fly (larvae) | 2.78 | 4.8 | 1.33 | 2.2 | 1.7 | 2.8 | 3.2 |
| *Littorina obtusata* | Periwinkle | 2.78 | 0.67 | 1.13 | 6.53 | 4.7 | 4.3 | 0.9 |
| Halacaridae (species 2) | Mite | 2.58 | 3.07 | 1.33 | 3.33 | 2.2 | 2.1 | 1.9 |
| *Nucella lapillus* | Dogwhelk | 2.18 | 0.67 | 5.33 | 0.53 | 4.1 | 0.6 | 4.0 |
| *Apohyale prevostii* | Amphipod | 1.53 | 3.73 | 0.6 | 0.27 | 0.6 | 2.7 | 2.9 |
| Copepoda (one species) | Copepods | 0.47 | 0.2 | 1.07 | 0.13 | 0.9 | 0.2 | 0.9 |
| *Hiatella arctica* | Bivalve mollusc | 0.47 | 1 | 0.4 | – | 0.3 | 0.8 | 1.0 |
| *Jaera albifrons* | Isopod | 0.33 | 0.73 | 0.13 | 0.13 | 0.2 | 0.5 | 0.6 |
| *Tetrastemma candidum* | Nemertean worm | 0.31 | – | – | 0.93 | 0.7 | 0.6 | <0.1 |
| *Foviella affinis* | Flatworm | 0.31 | 0.13 | 0.13 | 0.67 | 0.6 | 0.5 | 0.2 |
| *Testudinalia testudinalis* | Limpet | 0.29 | 0.87 | – | – | <0.1 | 0.8 | 0.9 |
| Nemertea (one species) | Nemertean worm | 0.24 | 0.2 | 0.53 | – | 0.5 | 0.1 | 0.5 |
| *Dynamena pumila* | Hydrozoan | 0.18 | 0.33 | 0.13 | 0.07 | 0.1 | 0.3 | 0.4 |
| *Eulalia viridis* | Polychaete | 0.16 | 0.33 | 0.13 | – | 0.1 | 0.3 | 0.4 |
| *Unciola serrata* | Amphipod | 0.09 | 0.27 | – | – | <0.1 | 0.2 | 0.2 |
| *Phyllodoce* sp. | Polychaete | 0.07 | 0.13 | 0.07 | – | <0.1 | 0.1 | 0.1 |
| *Onoba aculeus* | Snail | 0.07 | 0.07 | – | 0.13 | 0.1 | 0.1 | <0.1 |
| *Monoophorum* sp. | Flatworm | 0.07 | – | – | 0.2 | 0.2 | 0.2 | <0.1 |
| *Laomedea* sp. | Hydrozoan | 0.04 | – | 0.07 | 0.07 | 0.1 | <0.1 | <0.1 |
| *Anomia simplex* | Bivalve mollusc | 0.04 | 0.13 | – | – | <0.1 | 0.1 | 0.1 |
| *Littorina littorea* | Periwinkle | 0.04 | – | – | 0.13 | 0.1 | 0.1 | <0.1 |
| *Coronadena mutabilis* | Flatworm | 0.04 | 0.13 | – | – | <0.1 | 0.1 | 0.1 |
| *Cirratulus cirratus* | Polychaete | 0.02 | – | 0.07 | – | <0.1 | <0.1 | <0.1 |
| *Fabricia stellaris* | Polychaete | 0.02 | – | – | 0.07 | 0.1 | 0.1 | <0.1 |
| *Lepidonotus squamatus* | Polychaete | 0.02 | 0.07 | – | – | <0.1 | 0.1 | 0.1 |
| *Nereis* sp. | Polychaete | 0.02 | 0.07 | – | – | <0.1 | 0.1 | 0.1 |
| *Cerapus tubularis* | Amphipod | 0.02 | – | 0.07 | – | 0.1 | <0.1 | 0.1 |
| Hexacorallia (one species) | Anthozoans | 0.02 | 0.07 | – | – | <0.1 | <0.1 | <0.1 |
| *Sertularia cupressina* | Hydrozoan | 0.02 | – | 0.07 | – | <0.1 | <0.1 | <0.1 |

The most abundant species was an oligochaete (Tubificidae) that, although it contributed the most to explaining compositional differences between locations, was broadly abundant at all three locations (Table 1). The next most abundant species was the barnacle *Semibalanus balanoides*, but its abundance differed more markedly among locations,

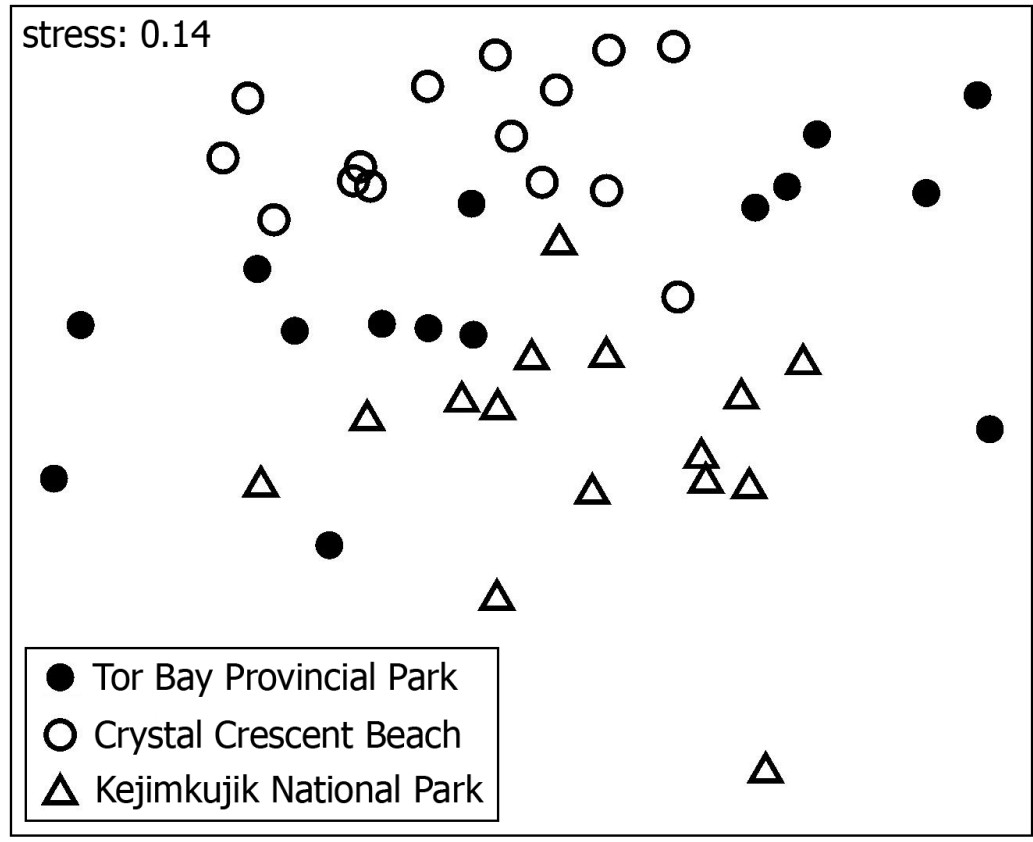

**Figure 3 Nonmetric multidimensional scaling (NMDS) ordination.** NMDS ordination of the mussel patches collected at three wave-exposed intertidal locations along the Nova Scotia coast based on the species composition of the associated invertebrate assemblages.

as it increased by nearly 200% from TB to KE and by nearly 1040% from CC to KE, on average (Table 1). All other species contributed considerably less to the compositional differences among locations, especially between KE and the two locations to the north. During April and the first half of May 2012 (the period most relevant for larvae explaining barnacle recruitment; see "Materials and methods"), chl-a in coastal waters was more than twice as high at KE (2.8 mg m$^{-3}$) than at TB (1.3 mg m$^{-3}$) and CC (1.1 mg m$^{-3}$).

## DISCUSSION

This study shows that invertebrate assemblages from mussel stands with a similar structure vary in species composition along the Nova Scotia coast. This is an important finding because it suggests that, while internal properties of mussel stands may remain relatively invariant along coastlines (*Jurgens & Gaylord, 2018*), the changes in external conditions that typically occur along coastlines (*Sanford, 2014*) might indeed influence the associated communities. These results thus align with those obtained for communities associated to other intertidal foundation species from other coasts (*Cole & McQuaid, 2011*; *Lloyd et al., 2020*).

Within biogeographic regions, community structure varies because of abiotic changes influencing species performance (environmental filtering), changes in interspecific interactions, and changes in food supply and recruitment (*Menge & Sutherland, 1987*; *Bruno, Stachowicz & Bertness, 2003*; *Vellend, 2016*). For rocky-intertidal communities of primary space holders and their consumers, there is a good understanding of how the above factors influence community structure for several coasts around the world (*Bertness, 2007*; *Benedetti-Cecchi & Trussell, 2014*; *Hawkins et al., 2019*; *Menge et al., 2019*). However, for the communities of small invertebrates living in stands of rocky-intertidal foundation species, an equivalent understanding is largely lacking. Knowledge gaps exist at almost all levels, from species physiological tolerances, to interspecific interactions, to recruitment within such stands. In addition, snapshot abundance data of the species composing a community often cannot unequivocally reveal the underlying interaction web (*Thurman et al., 2019*; *Blanchet, Cazelles & Gravel, 2020*), as interaction webs are shaped by direct and indirect interspecific interactions that are often unpredictable without experimentation (*Menge, 1995*). Therefore, below we briefly discuss possible reasons for the main observed patterns to orient future research (*Underwood, Chapman & Connell, 2000*).

After an oligochaete that was abundant at all three locations, barnacles were the second largest contributors to compositional differences among locations, peaking markedly in abundance at the southernmost location (KE). In 2014 (two years after these surveys), summer barnacle abundance on wave-exposed intertidal substrates outside of mussel stands was higher at two locations near KE than at TB and a location near CC. Summer barnacle abundance was then positively correlated to spring recruitment, which was in turn positively correlated to chl-a during the main larval period (*Scrosati & Ellrich, 2018*). Thus, it is possible that the higher chl-a recorded in spring 2012 at KE compared with TB and CC may have determined through increased recruitment the higher barnacle abundance found in mussel stands at KE in 2012. This suggests that benthic–pelagic coupling might influence the abundance of organisms living in stands of foundation species, a phenomenon that is already known to affect the abundance of primary space holders (*Navarrete et al., 2005*; *Menge et al., 2019*). Benthic–pelagic coupling might also have influenced the abundance of periwinkles in Nova Scotia's mussel stands. The two main encountered species, *Littorina saxatilis* and *L. obtusata*, showed their highest abundance also at KE (Table 1). During the growth period before the abundance measurements (March-August 2012), chl-a in coastal waters was twice as high at KE (4.4 mg m$^{-3}$) than at TB and CC (2.2 mg m$^{-3}$ at both locations). While chl-a reflects phytoplankton abundance, it also suggests that inorganic nutrients were likely in higher concentration at KE, which may have fueled a higher productivity of benthic microalgae that small periwinkles (those found in mussel stands) feed on. This possibility is supported by data indicating that upwelling (the upsurge of nutrient-rich waters) is more prevalent on southern shores in Nova Scotia during the growth season (*Scrosati & Ellrich, 2020*). Finally, there is also evidence suggesting that predator abundance might have been enhanced by prey abundance. For instance, oligochaetes were most abundant in mussel stands at the northernmost location (TB), which coincides with the highest combined abundance of the two recognized mite species

(Table 1). This pattern could reveal a case of bottom-up forcing (*Menge, 2000*) because intertidal mites are known to prey on oligochaetes (*Pugh & King, 1985*).

A side comment applies to species abundance in relation to species occurrence. While the species with the highest mean overall abundances occurred at all three studied locations, the species with low abundances typically occurred only at a subset of locations (Table 1). These patterns reflect basic abundance–occupancy patterns that are known for biological communities across landscapes and seascapes (*Blackburn, Cassey & Gaston, 2006*; *Barnes, 2022*; *van Genne & Scrosati, 2022*).

Overall, it is becoming clear that, like primary space holders, the small species associated to a given foundation species also vary in relative abundance along marine rocky shores. This study used species abundance data from mussel stands of the same structure (dense monolayered stands of small mussels from wave-exposed, mid-intertidal habitats) to exclude beforehand effects of stand structure on the associated communities. However, differences in intertidal elevation and wave exposure can alter structural properties of stands of foundation species, such as the density, size, and spatial arrangement of the individuals forming the stands (*Hammond & Griffiths, 2004*; *Tam & Scrosati, 2014*; *Vozzo et al., 2021*; *Wilbur, Küpper & Louca, 2024*). These changes, in turn, can alter the microclimates that affect the associated species (*McAfee et al., 2018*; *Veiga et al., 2022*). Thus, unraveling the drivers of overall alongshore changes in the communities of small species living in stands of foundation species will need to consider direct external influences on these associated species as well as possible structural changes in mussel stands that may influence local microclimates. This approach has recently been recognized as key to understand, in a broader context, the many contributions that communities associated to foundation species have to ecosystem stability and functioning (*De Frenne et al., 2021*).

It is worth noting that mussel stands are common in temperate rocky-intertidal habitats around the world and that, despite regional taxonomic differences rooted in biogeographic history, they host a core set of functional groups (*Cameron et al., 2024b*). Therefore, mussels are particularly relevant for the ecology of the coastal environments they inhabit. Intertidal stands of mussels, however, are increasingly being lost in relation to the ongoing climatic and oceanographic change (*Sorte et al., 2017*; *Seuront et al., 2019*; *Mendez et al., 2021*; *Fields & Silbiger, 2022*; *Raymond et al., 2022*; *Cameron & Scrosati, 2023*; *Scrosati, 2023*). This is, in fact, a trend seen also in other marine foundation species such as corals, seaweeds, seagrasses, salt marsh plants, mangroves, and other bivalves (*Wernberg et al., 2024*). For these reasons, knowledge that can be applicable to the conservation and restoration of these systems (*e.g.*, *Clausing et al., 2023*; *Toone et al., 2023*; *Benjamin et al., 2024*) is increasingly necessary.

## ACKNOWLEDGEMENTS

We are grateful to two anonymous reviewers for their constructive comments on an earlier version of this manuscript.

### Funding

This work was supported by a Discovery Grant (number 311624) awarded by the Natural Sciences and Engineering Research Council of Canada (NSERC) to Ricardo A. Scrosati. The funders had no role in study design, data collection and analysis, decision to publish, or preparation of the manuscript.

### Grant Disclosures

The following grant information was disclosed by the authors:
The Natural Sciences and Engineering Research Council of Canada (NSERC): 311624.

### Competing Interests

The authors declare there are no competing interests.

### Author Contributions

- Ricardo A. Scrosati conceived and designed the experiments, performed the experiments, analyzed the data, prepared figures and/or tables, authored or reviewed drafts of the article, and approved the final draft.
- Julius A. Ellrich performed the experiments, analyzed the data, authored or reviewed drafts of the article, and approved the final draft.

### Data Availability

The data is available at Scrosati, R. A., L. P. Arribas, and L. Donnarumma. 2020. Abundance data for invertebrate assemblages from intertidal mussel beds along the Atlantic Canadian coast. Ecology 101(10):e03137. https://doi.org/10.1002/ecy.3137.

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
