# Peer review of "Changes in the composition of invertebrate assemblages from wave-exposed intertidal mussel stands along the Nova Scotia coast, Canada"

_PeerJ, doi:10.7717/peerj.17697_

## Round 0.1 · original submission · Major Revisions

We have received evaluations of your manuscript from two expert reviewers and their comments can be seen below. As you will see that both raise important issues that need to be attended to in a revised manuscript accompanied by a rebuttal letter that clearly states what changes or modifications have been made and where in the manuscript. Please pay particular attention to the point raised by reviewer 1 about providing evidence for the invariant nature of the mussel stands throughout the region as well as the point raised by reviewer 2 about the novelty of this study with respect to that of Scrosati et al 2020.

Reviewer 1 ·

Basic reporting

Title: Two points here:
(i) I am not entirely sure about the “correctness” of using biogeographic in the title as the concept entails geographic differences as determined by the biota, which was not entirely the case. The mussel stands, the foundation species (as well as the most abundant associated species), were generally represent by the same species across the study region. In fact, this same point is recognised by the authors in the methods “These are cold-temperate locations located in the same hydrographic subregion (Nova Scotia) of the NW Atlantic cold-temperate biogeographic region”. Some replacements that may be a better fit could be geographic or spatial.

(ii) How relevant is it to have “wave-exposed” in the title? The way I see it, it is not that relevant and could be easily omitted to make the title more concise.

Line 38. Here and elsewhere (if relevant), consider replacing mobile consumers by motile consumers. According to the definition “Motility, the ability of an organism to move independently, using metabolic energy, can be contrasted with sessility, the state of organisms that do not possess a means of self-locomotion and are normally immobile. Motility differs from mobility, the ability of an object to be moved.”

Experimental design

Line 113. I have no problems with using ANOSIM, but PERMANOVA, which would likely show the same output, has two advantages: (i) it allows a posteriori comparisons (between pairs of locations) and, more importantly (ii) it allows estimating the components of variation associated with locations and replicate quadrats. This may further help understanding if variability in these assemblages is mostly associated with processes occurring at the scale of locations (e.g. such as differneces in Chl a among locations) or is mostly associated with processes occurring at much smaller (intra-location) scales (e.g. variation in density or size of mussel among patches).

Validity of the findings

This study examines the geographical variation in the structure of invertebrate assemblages associated with mussel stands. The study is nice and concise but one of the assumptions in way the manuscript is structured, and the underlying rationale is that mussel stands remain invariant across the study region. However, the authors fail to provide such evidence. Besides stating that samples were taken from areas where mussels form monolayered stands, there is no data supporting that there were no spatial differences in density and/or size of mussels among quadrats or locations. As foundation species, the spatial variation in the size or density of mussels may help explain some of the observed observation. As such, I believe this is a key point for this particular study as the way it is laid out and the hypothesis was built is centred in the fact that internal properties of the mussel stands remain invariant along the region. It is central to back up this and to have some measure of how variable (or invariable) mussels stands were along the study area.

As with barnacle recruitment, mussel recruitment, size or even mortality are likely to vary with coastal productivity. Otherwise, some claims throughout the manuscript i.e. “This study shows that invertebrate assemblages from mussel stands with a similar structure…”, “This study used species abundance data from mussel stands of the same structure (dense monolayered stands of small mussels from wave-exposed, mid-intertidal habitats) to exclude beforehand effects of stand structure on the associated communities.” need to be toned down, or the manuscript re-structured.

Additional comments

I believe that the key point raised by my review (that mussel stands actually remain invariant throughout the region) can be easily tackled by the authors provided that they have data to support it. (It is not clear if they do or do not have such data). Hence my suggestion of major revisions.

Reviewer 2 ·

Basic reporting

In this work authors explore invertebrate assemblages from wave-exposed intertidal mussel stands. It is an interesting work that present some insights on the interaction of organisms and their surroundings. Nevertheless, the novelty of the manuscript in relation to Scrosati et al. 2020 is not clear. What is the approach that differentiates this new proposed manuscript from the
Scrosati, R. A., Arribas, L. P., & Donnarumma, L. (2020). Abundance data for invertebrate assemblages from intertidal mussel beds along the Atlantic Canadian coast. Ecology, 101,e03137.
?
Additionally, authors talk about a benthic-pelagic coupling, but it is not clear to which couple they are referring to: is it barnacle-phytoplankton in a mussel strand? Moreover, barnacles tend to avoid settling on mussel shells, using rock or alternative substratum, if available. This should be further explored. What about the interface between mussels and barnacles? Are these interactions affecting other members of the community?
What the implications of the different assemblages are?
In general, the text needs some clearance.
Authors should avoid so many auto citations.
In more detail:
L56-57: The sentence needs to be simplified.
L76: The difference between the present study and Scrosati et al. (2020) should be stated clearly here.
L80-81: The meaning is not clear.
L110: The citation here seems that the data has already been published.
L150: “differences in the most abundant species”.
L167-169: Rather vague. More detail is needed.
L186-187: In relation to barnacles, data do not show a gradation between northern and southern geographic locations. The middle point has the lowest abundance.
L209-211: This statement needs more exploration due to the relevance of the data.
L214-217: Needs more detail. Was not this the goal? Assemblages comparation?
L222-225: Was this evident? Were there differences in the size of the mussels? Not according to L104-105.
L227-229: The meaning of the sentence is not clear.
Table 1: in the caption, in (individuals dm-2), “d” should be deleted. The name of sampling locations should be added to the acronyms in full.
“Common name” column should be changed to a phylum/class column.
Common names should be avoided along the text. “Seaweed” should be changed to “macroalgae”.

Experimental design

No comment.

Validity of the findings

No comment.

---

## Round 0.2 · Minor Revisions

Thank you for the modifications that you have made to the manuscript. The reviewers have a few minor points and I would like for you to consider then and either make modifications in a new version of the manuscript or justify why you did not make the suggested change. Please do this for the points mentioned by each reviewer.

Reviewer 1 ·

Basic reporting

I would consider replacing the word compositional by structural (in title and elsewhere).
Most ecologists tend to use the word compositional when referring to changes in species identities alone, disregarding abundance data (e.g. Jaccard). Whenever one refers to the combined measure of species identities and their relative abundances (e.g. Bray-Curtis, which was the case in this study), assemblage or community structure is more commonly used. I believe with this little revision, the wording of the manuscript will be more accurate for a general audience.

Experimental design

no comment

Validity of the findings

no comment

Additional comments

I am generally happy with the way the authors have revised their manuscript considering both mine and the other reviewer criticism/suggestions. I have no further comments.

Reviewer 2 ·

Basic reporting

The manuscript has improved from the first version.
There was still margin for adding some considerations on the interface between mussels and barnacles and how their interactions affect other members of the community, but it is not a crucial point.

Experimental design

No comment.

Validity of the findings

No comment.

---

## Round 0.3 · accepted · Accept

I am satisfied with the justification that the authors have supplied in response to the final minor comments from the reviewers.